# Improved rates for prediction and identification for partially observed linear dynamical systems

## Abstract

Identification of a linear time-invariant dynamical system from partial observations is a fundamental problem in control theory. Particularly challenging are systems exhibiting long-term memory. A natural question is how learn such systems with non-asymptotic statistical rates depending on the inherent dimensionality (order) $d$ of the system, rather than on the possibly much larger memory length. We propose an algorithm that given a single trajectory of length $T$ with gaussian observation noise, learns the system with a near-optimal rate of $\widetilde{O}\left(\sqrt{\frac{d}{T}}\right)$ in $\mathcal{H}_2$ error, with only logarithmic, rather than polynomial dependence on memory length. We also give bounds under process noise and improved bounds for learning a realization of the system. Our algorithm is based on multi-scale low-rank approximation: SVD applied to Hankel matrices of geometrically increasing sizes. Our analysis relies on careful application of concentration bounds on the Fourier domain – we give sharper concentration bounds for sample covariance of correlated inputs and for $\mathcal{H}_\infty$ norm estimation, which may be of independent interest.

## 1 Introduction

We consider the problem of prediction and identification of an *unknown* partially-observed linear time-invariant (LTI) dynamical system with stochastic noise,

$$x(t) = Ax(t-1) + Bu(t-1) + \xi(t) \tag{1}$$

$$y(t) = Cx(t) + Du(t) + \eta(t), \tag{2}$$

with a single trajectory of length $T$, given access only to input and output data. Here, $u(t) \in \mathbb{R}^{d_u}$ are inputs, $x(t) \in \mathbb{R}^d$ are the hidden states, $y(t) \in \mathbb{R}^{d_y}$ are observations (or outputs), $\xi(t) \sim N(0, \Sigma_x)$ and $\eta(t) \sim N(0, \Sigma_y)$ are iid gaussian noise, and $A \in \mathbb{R}^{d \times d}, B \in \mathbb{R}^{d \times d_u}, C \in \mathbb{R}^{d_y \times d}, D \in \mathbb{R}^{d_y \times d_u}$ are matrices. Partial observability refers to the fact that we do not observe the state $x(t)$, but rather a noisy linear observation $y(t)$.

As a simple and tractable family of dynamical systems, LTI systems are a central object of study for control theory and time series analysis. The problem of prediction and filtering for a known system dates back to [Kal60]. However, in many machine learning applications, the system is *unknown* and must be learned from input and output data. Identification of an unknown system is often a necessary first step for robust control [DMM+19, BMR18]. In a long line of recent work, the interplay between machine learning and control theory has borne fruit in an improved understanding of the statistical and online learning guarantees for prediction, identification, and control for unknown systems. In machine learning, LTI systems also serve as a simple model problem for learning from correlated

data in stateful environments, and can give insight into understanding the successes of reinforcement learning [Rec19, TR19] and recurrent neural networks [HMR18].

Partial observability poses a significant challenge to system identification: In the fully observed setting, given access to $x(t)$, there is no obstacle to learning the matrices directly through linear regression. However, in the partially observed setting, the most natural form of the optimization problem is non-convex.

Systems exhibiting *long-term memory* are particularly challenging to learn. Restricting to marginally stable systems, this occurs when the spectral radius of $A$, $\rho(A)$, is close to 1, and it implies that the output at a particular time cannot be accurately estimated without taking into account inputs over many previous time-steps—on the order of $O\left(\frac{1}{1-\rho(A)}\right)$ times steps. Such systems often arise in practice. A particular class of such systems are those exhibiting *multiscale* behavior, with different state variables that change on vastly different timescales [CR10]. For example, the body's pH level is affected both by long-term changes on a timescale of days or weeks, as well as breathing rate which changes over a timescale of seconds. For such systems, it makes sense to discretize at the scale of the fastest changing variable, which leads to a long memory for the slowest-changing variable. With few exceptions, existing guarantees for learning partially observed LTI systems degrade as the memory length increases. However, counting the number of parameters in the model (1)–(2) suggests that the right measure of statistical complexity is the intrinsic dimensionality of the system, not the memory length. This leads to the following natural question.

**Question:** How can we learn partially observed LTI systems with (non-asymptotic) statistical rates that depend on the *intrinsic dimensionality* of the system, rather than the memory length?

Despite the simplicity of the question, little in the way of theoretical results are known. We focus on the particular problem of learning the *impulse response (IR) function* of the system—which fully determines its input-output behavior—in $\mathcal{H}_2$ norm. This is a natural norm for prediction problems as it measures the expected prediction error under random input. Known guarantees for learning the IR depend on the memory length. One particularly undesirable consequence is that for a continuous system with time discretization $\Delta$ going to 0, the memory scales as $1/\Delta$ (while the system order stays constant), leading to suboptimal estimation by an arbitrarily large factor.

Our key contribution is an algorithm and analysis that gives statistical rates that are optimal up to logarithmic factors. Unlike previous works, our rates depend on the system order $d$—the natural dimensionality of the problem—and only *logarithmically* on the memory length of the system. Our algorithm is based on taking a low-rank approximation (SVD) of the Hankel matrix, which is a widely used technique in system identification. We consider a *multiscale* version of this algorithm, where we repeat this process for a geometric sequence of sizes of the Hankel matrix. This is essential for obtaining a stronger theoretical guarantee. In the setting of zero process noise, we prove that our algorithm achieves near-optimal $\widetilde{O}\left(\sqrt{\frac{d(d_u+d_y)}{T}}\right)$ rates in $\mathcal{H}_2$ error for the learned system.

Our analysis relies on careful application of concentration bounds on the Fourier domain to give sharper concentration bounds for sample covariance and $\mathcal{H}_\infty$ norm estimation, which may be of independent interest. While we consider our algorithm in a simple setting, we hope that this is a first step to understanding and improving more complex subspace identification algorithms. Indeed, SVD and related spectral methods are a standard step used in subspace identification algorithms such as N4SID; our analysis suggests that SVD has an important "de-noising effect".

We also give improved bounds for system identification, that is, learning the matrices $A, B, C, D$ using the Ho-Kalman algorithm [HK66], with $\widetilde{O}\left(\sqrt{\frac{Ld(d_u+d_y)}{T}}\right)$ rates.

## 1.1 Notation

**Norms.** We use $\|\cdot\|$ to denote the 2-norm of a vector. For a matrix $A$, let $\|A\| = \|A\|_2$ denote its operator norm, $\rho(A)$ denote its spectral radius (maximum absolute value of eigen-

78 value), and $\|A\|_{\mathsf{F}}$ denote its Frobenius norm. For a matrix-valued function $M(t) \in \mathbb{C}^{d_1 \times d_2}$,
79 $\|M\|_{\mathsf{F}} := \sqrt{\sum_t \|M(t)\|_{\mathsf{F}}^2}$. Let $\sigma_r(A)$ denote the $r$th singular value of $A$.

**Fourier transform.** Given a matrix-valued function $F : \mathbb{Z} \to \mathbb{C}^{m \times n}$, define the (discrete-time)
81 Fourier transform as the function $\widehat{F} : \mathbb{R}/\mathbb{Z} \to \mathbb{C}^{m \times n}$ given by $\widehat{F}(\omega) = \sum_{t=-\infty}^{\infty} F(t)e^{-2\pi i \omega t}$.

**Matrices.** Given a sequence $(F(t))_{t=1}^{a+b-1}$ where each $F(t) \in \mathbb{C}^{m \times n}$, define $\mathrm{Hankel}_{a \times b}(F)$ as the
83 $am \times bn$ block matrix such that the $(i, j)$th block is $[\mathrm{Hankel}_{a \times b}(F)]_{ij} = F(i + j - 1)$. Given a
84 sequence $(F(t))_{t=0}^{a-1}$ where each $F(t) \in \mathbb{C}^{m \times n}$, define the Toeplitz matrix as the block matrix such
85 that the $(i, j)$th block is $[\mathrm{Toep}_{a \times b}(F)]_{ij} = F(i - j)\mathbb{1}_{i \geq j}$. For a matrix $A$, let $A^\top, A^H, A^\dagger$ denote
86 its transpose, Hermitian (conjugate transpose), and pseudoinverse, respectively. For a vector-valued
87 function $v : \{a, \ldots, b\} \to \mathbb{R}^n$, let $v_{a:b} \in \mathbb{R}^{(|a-b|+1)n}$ denote the the vertical concatenation of
88 $v(a), \ldots, v(b)$.

**Control theory.** For a matrix $A \in \mathbb{C}^{d \times d}$, define its resolvent as $\Phi_A(z) = (zI - A)^{-1}$. For a
90 linear dynamical system $\mathcal{D}$ given by (1)–(2), let $\Phi_{\mathcal{D}} = \Phi_{u \to y}$ denote the transfer function from
91 $u$ to $y$ (response to input). Then $\Phi_{\mathcal{D}} = \Phi_{u \to y} = C\Phi_A B + D = C(zI - A)^{-1}B + D$. Let
92 $\mathbb{T} := \{z \in \mathbb{C} : |z| = 1\}$ be the unit circle in the complex plane. For a matrix-valued function $F :$
93 $\mathbb{T} \to \mathbb{C}^{d_1 \times d_2}$, define the $\mathcal{H}_2$ and $\mathcal{H}_\infty$ norms by

$$\|\Phi\|_{\mathcal{H}_2} = \sqrt{\frac{1}{2\pi} \int_{\mathbb{T}} \|\Phi(z)\|_F^2 \ dz} \qquad\qquad \|\Phi\|_{\mathcal{H}_\infty} = \sup_{z \in \mathbb{T}} \|\Phi(z)\| \, .$$

94 For a function $F : \mathbb{N}_0 \to \mathbb{C}^{d_1 \times d_2}$, define its Z-transform to be $\mathcal{Z}[F](z) = \sum_{n=0}^{\infty} F(n)z^{-n}$. Con-
95 sidered as a function $\mathbb{T} \to \mathbb{C}$, we can take its $\mathcal{H}_2$ and $\mathcal{H}_\infty$ norms. Overloading notation, we will let
96 $\|F\|_{\mathcal{H}_p} := \|\mathcal{Z}F\|_{\mathcal{H}_p}$ for $p = 2, \infty$. The $\mathcal{H}_2$ and $\mathcal{H}_\infty$ norms can be interpreted as the Frobenius and
97 operator norms of the linear operator from input to output, i.e., they measure the average power of
98 the output signal under random or worst-case input, respectively. For background on control theory,
99 see e.g., [ZDG$^+$96].

**Constants.** In proofs, $C$ may represent different constants from line to line.

## 2 Main results

102 We consider the problem of prediction and identification for an unknown linear dynamical sys-
103 tem (1)–(2). Our main goal is to obtain error guarantees in $\mathcal{H}_2$ norm, which determines prediction
104 error under random input [OO19, Lemma 3.3].

**Problem 2.1.** *Consider the partially-observed LTI system $\mathcal{D}$ (1)–(2) with gaussian inputs $u(t) \sim$*
106 *$N(0, I_{d_u})$ for $0 \leq t < T$. Suppose that the system is stable, that is, $\rho(A) < 1$, and that we observe*
107 *a single trajectory of length $T$ started with $x(0) = 0$, that is, we observe $u(t) \sim N(0, I_{d_u})$ and $y(t)$*
108 *for $0 \leq t < T$.*

*The goal is to learn a LTI system $\widetilde{\mathcal{D}}$ with the aim of minimizing $\left\|\Phi_{\widetilde{\mathcal{D}}} - \Phi_{\mathcal{D}}\right\|_{\mathcal{H}_2}$. Equivalently, letting*

$$F^*(t) = \begin{cases} D, & t = 0 \\ CA^{t-1}B, & t \geq 1 \end{cases}$$

110 *denote the impulse response function (also called the Markov parameters) of the system, the goal is*
111 *to learn an impulse response $\widetilde{F}$ minimizing $\left\|F^* - \widetilde{F}\right\|_{\mathcal{H}_2} = \left\|F^* - \widetilde{F}\right\|_{\mathsf{F}}$.*

112 Note that learning $F^*$ is sufficient to fully understand the input-output behavior of the system, but
113 we may also ask to recover the system parameters $A, B, C, D$ up to similarity transformation (see
114 Theorem 2.3).

115 Previous results [OO19, SRD19] roughly depend polynomially on the "memory" $\frac{1}{1-\rho(A)}$, which
116 blows up as the spectral norm of $A$ approaches 1. In the setting of zero process noise, our goal is

to obtain rates that are $\widetilde{O}\left(\frac{\text{poly}(d,d_u,d_y)}{\sqrt{T}}\right)$, with only poly-logarithmic dependence on $\frac{1}{1-\rho(A)}$. See Figure 1 for a comparison.

We assume that $\rho(A) < 1$ because if $\mathcal{D}$ is not stable, it is in general impossible to learn $\widetilde{\mathcal{D}}$ with finite $\mathcal{H}_2$ error, as a system with infinite response can have arbitrarily small response on any finite time interval. However, it may still be possible to learn the response up to time $L \ll T$ in this case [SBR19]. The marginally stable case ($\rho(A) = 1$) is an important case we leave to future work.

| Method | Rollout type | Min # samples | IR error |
|---|---|---|---|
| Least squares (IR) [TBPR17] | Multi | $L$ | $\sigma\sqrt{\frac{L}{T}}$ |
| Least squares (IR) [OO19] | Single | $L$ | $\sigma\sqrt{\frac{L}{T}}$ |
| Nuclear norm minimization | Multi | $\min\{d^2, L\}$ | $\sigma\sqrt{\frac{L}{T}}$ |
| [SOF20] | Multi | $d$ | $\sigma\sqrt{\frac{dL}{T}}$ |
| rank-$d$ SVD (Theorem 2.2) | Single | $L$ | $\sigma\sqrt{\frac{d}{T}}$ |

Figure 1: Here, $L$ is the memory length for the system, which is $\widetilde{O}\left(\frac{1}{1-\rho(A)}\right)$ for well-conditioned systems. *Rollout type* refers to whether we have access to a single trajectory or multiple trajectories. *Min # samples* refers to the minimum number of samples (up to log factors) before the bounds are operational. *IR error* refers to the error in the impulse response in Frobenius/$\mathcal{H}_2$ norm. Logarithmic factors are omitted.

In our Algorithm 1, we first use linear regression to obtain a noisy estimate $F$ of the impulse response. Next, following standard system identification procedures, we form the Hankel matrix $\text{Hankel}_{L \times L}(F)$ with the entries of $F$ on its diagonals. Because the true Hankel matrix

$$\text{Hankel}_{L \times L}(F^*) = \begin{pmatrix} CB & CAB & \cdots & CA^{L-1}B \\ CAB & CA^2B & & \vdots \\ \vdots & & \ddots & \vdots \\ CA^{L-1}B & \cdots & \cdots & CA^{2L-1}B \end{pmatrix}$$

has rank $d$, we take a low-rank SVD $R_L$ of the Hankel matrix to "de-noise" the impulse response. We can then read off the estimated impulse response by averaging over the corresponding diagonal of $R_L$. For technical reasons, we need to repeat this process for a geometric sequence of sizes of the Hankel matrix: $L \times L$, $L/2 \times L/2$, $L/4 \times L/4$, and so forth. This is because the low-rank approximation objective for a $\ell \times \ell$ Hankel matrix encourages the diagonals that are $\Theta(\ell)$ to be close—as those are the diagonals with the most entries—and hence estimates $F^*(t)$ well when $t = \Theta(\ell)$. In other words, low-rank estimation for $\text{Hankel}_{\ell \times \ell}(F)$ is only sensitive to the portion of the signal that is at timescale $\ell$. Repeating this process ensures that we cover all scales.

Our main theorem is the following.

**Theorem 2.2.** *There is a constant $C_1$ such that following holds. In the setting of Problem 2.1, suppose that $F^*$ is the impulse response function, $T$ is such that $T \geq C_1 L d_u \log\left(\frac{L d_u}{\delta}\right)$, $\varepsilon_{\text{trunc}} := \left\|F^* \mathbb{1}_{[2L,\infty)}\right\|_{\mathcal{H}_\infty} \sqrt{d_u} + \left\|G^* \mathbb{1}_{[2L,\infty)}\right\|_{\mathcal{H}_\infty} \left\|\Sigma_x^{1/2}\right\|_{\mathsf{F}}$, and $M_{x \to y} = (O, C, CA, \ldots, CA^{L-1})^\top \in \mathbb{R}^{(L+1)d \times d_y}$. Let $0 < \delta \leq \frac{1}{2}$ and $\sigma = \sqrt{\|\Sigma_y\| + \|\Sigma_x\| L \log\left(\frac{L d_u}{\delta}\right) \|M_{x \to y}\|^2}$. Then with probability at least $1 - \delta$, Algorithm 1 learns an impulse response function $\widetilde{F}$ such that*

$$\left\|\widetilde{F} - F^*\right\|_{\mathsf{F}} = O\left(\sigma\sqrt{\frac{d\left(d_y + d_u + \log\left(\frac{L}{\delta}\right)\right)\log L}{T}} + \varepsilon_{\text{trunc}}\sqrt{d} + \left\|F^* \mathbb{1}_{(L,\infty)}\right\|_{\mathsf{F}}\right).$$

In the absence of process noise (when $\Sigma_x = O$), when $L$ and $T$ are chosen large enough, the first term dominates, and ignoring log factors, the dependence is $O\left(\sqrt{\frac{d(d_y + d_u)}{T}}\right)$. We expect this to be

---

**Algorithm 1** Learning impulse response through multi-scale low-rank Hankel SVD

---

1: **Input:** Length $L$ (power of 2), time $T$.
2: Part 1: Linear regression to recover noisy impulse response
3: Let $u(t) \sim N(0, I_{d_u})$ for $0 \le t < T$, and observe the outputs $y(t) \in \mathbb{R}^{d_y}, 0 \le t < T$.
4: Solve the least squares problem

$$\min_{F:\mathrm{Supp}(F) \subseteq [0, 2L-1]} \sum_{t=0}^{T-1} \|y(t) - F * u(t)\|^2.$$

to obtain the noisy impulse response $F : [0, 2L-1] \cap \mathbb{Z} \to \mathbb{R}^{d_y \times d_u}$.
5: Part 2: Low-rank Hankel SVD to de-noise impulse response
6: Let $\widetilde{F}(0) = F(0)$.
7: **for** $k = 0$ to $\log_2 L$ **do**
8:     Let $\ell = 2^k$.
9:     Let $R_\ell$ be the rank-$d$ SVD of $\mathrm{Hankel}_{\ell \times \ell}(F)$ (i.e., $\mathrm{argmin}_{\mathrm{rank}(R) \le d} \|R - \mathrm{Hankel}_{\ell \times \ell}(F)\|$).
10:     For $\frac{\ell}{2} < t \le \ell$, let $\widetilde{F}(t)$ be the $d_y \times d_u$ matrix given by $\widetilde{F}(t) = \frac{1}{t} \sum_{i+j=t} (R_\ell)_{ij}$, where $(\cdot)_{ij}$ denotes the $(i, j)$th block of the matrix.
11: **end for**
12: **Output:** Estimate of impulse response $\widetilde{F}$.

---

the optimal sample complexity up to logarithmic factors. However, in the presence of process noise, there is an undesirable factor of $\sqrt{L} \|M_{x \to y}\|$, which (for well-conditioned matrices) is expected to be $O\left(\frac{1}{1-\rho(A)}\right)$ or $O(L)$. We leave it an open problem to improve the guarantees in this setting.

*Remark 1.* The $L$-factor dependence on the process noise is unavoidable with the current algorithm: when the process noise has covariance $\Sigma_y = I$ and decays after $L$ steps, it can cause perturbations of size $O(\sqrt{L})$ compared to the noiseless system. Even in the case $d = 1$, when the impulse response function is $ae^{-kt/L}$ for a known $k$, the noise will cause the estimate of $a$ to be off by $O(\sqrt{L})$, and hence the $\mathcal{H}_2$ norm of the impulse response to be off by $O(L)$. Our algorithm only regresses on previous inputs, but in the presence of process noise, a better approach is to regress on both the previous inputs $u(t)$ and *outputs* $y(t)$ and then take a (weighted) SVD, as in N4SID [Qin06].

*Remark 2.* A burn-in time of $\Omega(L)$ is information-theoretically required to get $\mathrm{poly}(d)$ rates. Attempting to extrapolate an impulse response function from time $o(L)$ to time $L$ can magnify errors by $\exp(d)$, because the finite impulse response of a system of order $d$ can approximate a polynomial of degree $d - 1$ on $[0, L]$.

We also show the following improved rates for learning the system matrices, by combining $\mathcal{H}_\infty$ bounds for the learned impulse response with stability results for the Ho-Kalman algorithm [OO19]. Because the input-output behavior is unchanged under a similarity transformation $(A, B, C) \hookleftarrow (W^{-1}AW, W^{-1}B, CW)$, we can only learn the parameters up to similarity transformation.

**Theorem 2.3.** *Keep the assumptions and notation of Theorem 2.2, suppose $\mathcal{D}$ is observable and controllable, and let*

$$\varepsilon' = \sigma \sqrt{\frac{L \left(d_y + d_u + \log\left(\frac{L}{\delta}\right)\right)}{T}} + \varepsilon_{\mathrm{trunc}}.$$

*Let $H^- = \mathrm{Hankel}_{L \times (L-1)}(F^*)$. Suppose that $\varepsilon' = O(\sigma_{\min}(H^-))$. Then with probability at least $1 - \delta$, the Ho-Kalman algorithm (Algorithm 1 in [OO19] with $T_1 = L, T_2 = L-1$) returns $\widehat{A}, \widehat{B}, \widehat{C}$ such that there exists a unitary matrix $W$ satisfying*

$$\max\left\{\left\|C - \widehat{C}W\right\|_{\mathsf{F}}, \left\|B - W^{-1}\widehat{B}\right\|_{\mathsf{F}}\right\} = O(\sqrt{d} \cdot \varepsilon')$$

$$\left\|A - W^{-1}\widehat{A}W\right\|_{\mathsf{F}} = O\left(\frac{1}{\sigma_{\min}(H^-)} \cdot \sqrt{d} \cdot \varepsilon' \cdot \left(\frac{\|\Phi_{\mathcal{D}}\|_{\mathcal{H}_\infty}}{\sigma_{\min}(H^-)} + 1\right)\right).$$

165 As $L$ can be chosen to make $\varepsilon_{\text{trunc}}$ negligible, this gives $\widetilde{O}\left(\sqrt{\frac{Ld(d_u+d_y)}{T}}\right)$ rates, however,
166 with factors depending on the minimum eigenvalue of $H$. This is an improvement over the
167 $\widetilde{O}\left(\sqrt{d}\sqrt[4]{\frac{L(d_u+d_y)}{T}}\right)$ rates in [OO19].

168 We prove Theorem 2.2 in Section 4 and Theorem 2.3 in Appendix B.

# 3 Related work

170 We survey two classes of methods for learning partially observable LDS's, subspace identification
171 and improper learning. With the exception of [RJR20], all guarantees have sample complexity
172 depending on the memory length $L$, which we wish to avoid.

## 3.1 Subspace identification

174 The basic idea of subspace identification [Lju98, Qin06, VODM12] is to learn a certain structured
175 matrix (such as a Hankel matrix), take a best rank-$k$ approximation (using SVD or another linear di-
176 mensionality reduction method), and learn the system matrices $A, B, C, D$ up to similarity transfor-
177 mation. Usage of spectral methods circumvents the fact that the most natural optimization problem
178 for $A, B, C, D$ is non-convex. However, classical guarantees for these methods are asymptotic.

179 Recently, various authors have given non-asymptotic guarantees for system identification algo-
180 rithms. [OO19] analyzed the Ho-Kalman algorithm [HK66] in this setting. [SRD19] consider the
181 setting where system order is unknown and give an end-to-end result for prediction, while [TMP20]
182 consider the problem of online filtering, that is, recovering $x(t)$'s up to some linear transformation.

183 An alternate, empirically successful approach is that of nuclear norm minimization or regulariza-
184 tion [FPST13]. [SOF20] (building on [CQXY16]) give explicit rates of convergence, and show that
185 the algorithm has a lower minimum sample complexity and is easier to tune.

186 Our algorithm is based on the classical approach of taking a low-rank approximation of the Hankel
187 matrix, but we repeat this process with Hankel matrices of sizes $L \times L$, $L/2 \times L/2$, $L/4 \times L/4$, and so
188 forth; this is key modification that allows us to obtain better statistical rates. Our analysis builds on
189 the analyses given in [OO19, SOF20]. As essential part of the analysis is analyzing linear regression
190 for correlated inputs, where we extend the work of [DMR19] to MIMO systems, as explained below.

### 3.1.1 Linear regression with correlated inputs

192 An important step in obtaining non-asymptotic rates for system identification is analyzing linear
193 regression for correlated inputs. The most challenging step is to lower-bound the sample covariance
194 matrix of inputs to the linear regression. A lower bound, rather than a matrix concentration result, is
195 sufficient [Men14, SMT$^+$18, MT19]; however, a concentration result is obtainable in our setting.

196 [TBPR17] give non-asymptotic bounds for learning the finite impulse response for a SISO system
197 in $\ell^\infty$ Fourier norm; however, they require $L$ rollouts of size $O(L)$ and hence $\Omega(L^2)$ timesteps.
198 Addressing the more challenging single-rollout setting, [OO19] obtain bounds for a single rollout
199 of $\widetilde{\Omega}(L)$ timesteps, by using concentration bounds for random circulant matrices [KMR14] to de-
200 rive concentration inequalities for the covariance matrix. These concentration inequalities for the
201 covariance matrix were improved (by logarithmic factors) by [DMR19]. Although [DMR19] give
202 an analysis in the SISO case, as we show in Theorem A.2, the results can be extended to the MIMO
203 case with an $\varepsilon$-net argument.

## 3.2 Improper learning using autoregressive methods

205 Instead of solving the statistical problem of identifying parameters, another line of work develops
206 algorithms for regret minimization in online learning. The goal is simply to do well in predicting
207 future observations, with small loss (regret) compared to the best predictor in hindsight; the learned
208 predictor is allowed to be improper, that is, take a different functional form. In the stochastic case,

209 this allows prediction almost as well as if the actual system parameters were known; however, the
210 framework also allows for adversarial noise.

211 One popular strategy for improperly learning the system is to learn a linear autoregressive filter over
212 previous inputs and observations, or ARMA model. Naturally, because we are optimizing over a
213 larger hypothesis class, the statistical rates depend on $L$ rather than the system order $d$.

214 [GLS$^+$20, Theorem 4.7] consider the problem of online prediction for a fully or partially observed
215 LDS, and give a regret bound that depends polynomially on the memory length $L$. Their approach
216 works even for marginally stable systems, that is, systems with $\rho(A) \leq 1$. See also [AHMS13,
217 HSZ17, HLS$^+$18, KMTM19, TP20, RJR20] for previous work using autoregressive methods.

218 Of particular interest to us is [RJR20], which gives rates independent of spectral radius. Building
219 on [HSZ17], they observe that it suffices to regress on previous inputs and outputs projected to a
220 lower-dimensional space. Their algorithm works in the setting of process noise and competes with
221 the Kalman filter, but only when $A - KC$ has real eigenvalues, where $K$ is the Kalman gain.

## 4 Proof of main theorem

223 In this section, we prove Theorem 2.2. The proof hinges on the following lemma, which shows
224 that if we observe a low-rank matrix plus noise, then taking a low-rank SVD can have a de-noising
225 effect, producing a matrix that is closer to the true matrix.

226 **Lemma 4.1** (De-noising effect of SVD). *There exists a constant $C$ such that the following holds.*
227 *Suppose that $A \in \mathbb{C}^{m \times n}$ is a rank-$r$ matrix, $\widehat{A} = A + E$, and $\widehat{A}_r$ is the rank-$r$ SVD of $\widehat{A}$. Then*

$$\left\| \widehat{A}_r - A \right\|_{\mathsf{F}} \leq C\sqrt{r} \left\| E \right\|. \tag{3}$$

228 Compare this with the original error $\left\| \widehat{A} - A \right\|_{\mathsf{F}} = \left\| E \right\|_{\mathsf{F}}$, which can only be bounded by
229 $\sqrt{\min\{m, n\}} \left\| E \right\|$. When applied to the $d$-SVD of the Hankel matrix, this gives a factor of $\sqrt{d}$
230 rather than $\sqrt{L}$ for the error.

231 *Proof.* We have

$$\left\| \hat{A}_r - A \right\|_{\mathsf{F}} \leq \sqrt{2r} \left\| \hat{A}_r - A \right\|_2 \tag{4}$$

$$\leq \sqrt{2r} \left( \left\| \hat{A}_r - \hat{A} \right\|_2 + \left\| \hat{A} - A \right\|_2 \right) \tag{5}$$

$$\leq 2\sqrt{2r} \left\| E \right\| \tag{6}$$

232 where (4) follows from $\hat{A}_r - A$ having rank at most $2r$, (5) follows from the triangle inequality,
233 and (6) follows from Weyl's Theorem: $\left\| \hat{A}_r - \hat{A} \right\|_2 \leq \sigma_{r+1}(\hat{A}) \leq \sigma_{r+1}(A) + \left\| E \right\| = \left\| E \right\|$. $\square$

234 To prove Theorem 2.2, we will need to obtain bounds for $F : \{0, 1, \dots, 2L-1\} \to \mathbb{R}^{d_y \times d_u}$ learned
235 from linear regression in $\mathcal{H}_\infty$ norm. The following is our main technical result.

236 **Lemma 4.2.** *There are $C_1, C_2$ such that the following hold. Suppose $y = F^* * u + G^* *$*
237 *$\xi + \eta$ where $u(t) \sim N(0, I_{d_u})$, $\xi(t) \sim N(0, \Sigma_x)$, $\eta(t) \sim N(0, \Sigma_y)$ for $0 \leq t < T$,*
238 *and $\mathrm{Supp}(F^*), \mathrm{Supp}(G^*) \subseteq [0, \infty)$. Let $F = \mathrm{argmin}_{F \in \{0,\dots,L\} \to \mathbb{R}^{d_y \times d_u}} \sum_{t=0}^{T-1} |y(t) - (F *$*
239 *$u)(t)|^2$, $M_{G^*} = (G^*(0), \dots, G^*(L))^\top \in \mathbb{R}^{(L+1)d \times d_y}$, and $\varepsilon_{\mathrm{trunc}} = \left\| F^* \mathbb{1}_{[L+1, \infty)} \right\|_{\mathcal{H}_\infty} \sqrt{d_u} +$*
240 *$\left\| G^* \mathbb{1}_{[L+1, \infty)} \right\|_{\mathcal{H}_\infty} \left\| \Sigma_x^{1/2} \right\|_{\mathsf{F}}$. For $0 < \delta \leq \frac{1}{2}$, $T \geq C_1 L d_u \log\left(\frac{L d_u}{\delta}\right)$, $-1 \leq a < L - L'$,*

$$\left\| (F - F^*) \mathbb{1}_{[a+1, a+L']} \right\|_{\mathcal{H}_\infty}$$

$$\leq C_2 \left[ \sqrt{\frac{1}{T}} \left( \sqrt{\left\| \Sigma_y \right\| L' \left( d_u + d_y + \log\left(\frac{L'}{\delta}\right) \right)} + \sqrt{\left\| \Sigma_x \right\| L' L d_u \log\left(\frac{L d_u}{\delta}\right)} \left\| M_{G^*} \right\| \right) + \varepsilon_{\mathrm{trunc}} \right]$$

241 *with probability at least $1 - \delta$.*

In the case $\Sigma_x = O$, this roughly says that the error in the learned impulse response, $F - F^*$, over any interval of length $L'$, has all Fourier coefficients bounded in spectral norm by $\widetilde{O}\left(\sqrt{\frac{L'(d_u + d_y)}{T}}\right)$ – what we expect if the error from linear regression is uniformly distributed over all frequencies.

A complete proof is in Appendix A; we give a brief sketch. First, because the errors are Gaussian, the error from linear regression, $F - F^*$, follows a Gaussian distribution. To bound its covariance, we lower-bound the smallest singular value of the sample covariance of the inputs (Lemma A.1, Appendix A.1). Here, the difficulty is that the inputs are *correlated* – the input at time $t$ is $u_{t:t-L}$. Fortunately, the translation structure means it is close to a submatrix of an infinite block Toeplitz matrix, which becomes block diagonal in the Fourier domain. This "decoupling" allows us to show concentration. Compared to the SISO setting in [DMR19], we require an extra $\varepsilon$-net argument. Once we have a bound on the covariance, we can bound any $\left\|\widehat{(F - F^*)}(\omega)\right\|$ by matrix concentration (Appendix A.2); to bound the $\mathcal{H}_\infty$ norm it suffices to bound this over a grid of $\omega$'s (Lemma A.4).

Bounding the error in $\mathcal{H}_\infty$ norm of the impulse response allows us to bound the error in operator norm of the Hankel matrix, as the following lemma shows.

**Lemma 4.3.** *For any $F : \mathbb{Z} \to \mathbb{C}^{m \times n}$, we have $\|\mathrm{Hankel}_{a \times b}(F)\| \leq \|F\|_{\mathcal{H}_\infty}$.*

*Proof.* Note that when $v : \mathbb{Z} \to \mathbb{C}^n$, $\mathrm{Supp}(v) \subseteq [0, b-1]$, we have $(F * v)_{b:b+a-1} = \mathrm{Hankel}_{a \times b}(F)v_{b-1:0}$. Hence for any $v$ set to 0 outside of $[0, b-1]$, using Parseval's Theorem and the fact that the Fourier transform of a convolution is the product of the Fourier transforms, we have

$$\|\mathrm{Hankel}_{a \times b}(F)v_{b-1:0}\|_2 \leq \|F * v\|_2 = \left\|\widehat{F}\widehat{v}\right\|_2 \leq \sup_{\omega \in [0,1]} \left\|\widehat{F}(\omega)\right\|_2 \|\widehat{v}\|_2 = \|F\|_{\mathcal{H}_\infty} \|v\|_2$$

This shows that $\|\mathrm{Hankel}_{a \times b}(F)\| \leq \|F\|_{\mathcal{H}_\infty}$. $\qquad\square$

Theorem 2.2 will follow from the following bound after an application of the triangle inequality.

**Lemma 4.4.** *There are $C_1, C_2$ such that the following holds for the setting of Problem 2.1. Suppose $L$ is a power of 2, and $T \geq C_1 L d_u \log\left(\frac{L d_u}{\delta}\right)$. Let $\left\|F^* \mathbb{1}_{[L+1,\infty)}\right\|_{\mathcal{H}_\infty} \sqrt{d_u} + \left\|G^* \mathbb{1}_{[L+1,\infty)}\right\|_{\mathcal{H}_\infty} \left\|\Sigma_x^{1/2}\right\|_{\mathsf{F}}$ and $M_{x \to y} = (O, C, CA, \ldots, CA^{L-1})^\top \in \mathbb{R}^{(L+1)d \times d_y}$. Then with probability at least $1 - \delta$, the output $\widetilde{F}$ given by Algorithm 1 satisfies*

$$\left\|(\widetilde{F} - F^*)\mathbb{1}_{[1,L]}\right\|_{\mathsf{F}} \leq C_2 \left(\sqrt{\frac{\|\Sigma_y\| d (d_y + d_u + \log(\frac{L}{\delta})) \log L}{T}} + \sqrt{\frac{\|\Sigma_x\| L d d_u \log(\frac{L d_u}{\delta})}{T}} \|M_{G^*}\| + \varepsilon_{\mathrm{trunc}} \sqrt{d}\right)$$

*Proof.* We are in the situation of Lemma 4.2 with $G^*(t) = CA^{t-1}\mathbb{1}_{t \geq 1}$. Let $\mathcal{H}_\ell = \mathrm{Hankel}_{\ell \times \ell}(F)$ and $\mathcal{H}_\ell^* = \mathrm{Hankel}_{\ell \times \ell}(F^*)$. Suppose $\ell \leq L$ is even. Note that

$$\mathcal{H}_\ell = \underbrace{\mathrm{Hankel}_{\ell \times \ell}(F^*)}_{\mathcal{H}_\ell^*} + \mathrm{Hankel}_{\ell \times \ell}(F - F^*)$$

where $\mathcal{H}_\ell^* = \mathrm{Hankel}_{\ell \times \ell}(F^*)$ is a rank-$d$ matrix, with error term is bounded by

$$\|\mathrm{Hankel}_{\ell \times \ell}(F - F^*)\| \leq \left\|(F - F_{\mathrm{trunc}}^*)\mathbb{1}_{[1,2\ell-1]}\right\|_{\mathcal{H}_\infty} \qquad \text{by Lemma 4.3}$$

$$< C \left[\sqrt{\frac{1}{T}}\left(\sqrt{\|\Sigma_y\| \ell \left(d_u + d_y + \log\left(\frac{L'}{\delta}\right)\right)}\right) \right.$$

$$\left. + \sqrt{\|\Sigma_x\| \ell L d_u \log\left(\frac{L d_u}{\delta}\right)} \|M_{G^*}\| + \varepsilon_{\mathrm{trunc}}\right] \qquad \text{by Lemma 4.2} \qquad (7)$$

with probability at least $1 - \delta$. Let $R_\ell$ be the rank-$d$ SVD of $\mathcal{H}_\ell$. Then by Lemma 4.1,

$$\|R_\ell - \mathcal{H}_\ell^*\|_{\mathsf{F}} = O\left(\sqrt{d} \|\mathrm{Hankel}_{\ell \times \ell}(F - F^*)\|\right). \qquad (8)$$

Now letting $\widetilde{F}(t) = \frac{1}{t} \sum_{i+j=t} (R_\ell)_{ij}$ when $\frac{\ell}{2} < t \leq \ell$ we have (using the fact that the mean minimizes the sum of squared errors)

$$\|R_\ell - \mathcal{H}_\ell^*\|_{\mathsf{F}}^2 \geq \sum_{t=\frac{\ell}{2}+1}^{\ell} \sum_{i+j=t} \|(R_\ell)_{ij} - F^*(t)\|_{\mathsf{F}}^2$$

$$\geq \sum_{t=\frac{\ell}{2}+1}^{\ell} \left( t \cdot \left\| \widetilde{F}(t) - F^*(t) \right\|_{\mathsf{F}}^2 \right) \geq \left( \frac{\ell}{2} + 1 \right) \sum_{t=\frac{\ell}{2}+1}^{\ell} \left( \left\| \widetilde{F}(t) - F^*(t) \right\|_{\mathsf{F}}^2 \right).$$

Note that we only get a lower bound with a factor of $\ell$ if we restrict to $t$ that is $\Theta(\ell)$, i.e., restrict to diagonals that have many entries. This is the reason we will have to repeat this process for multiple sizes. Hence

$$\left\| (\widetilde{F} - F^*) \mathbb{1}_{[\frac{\ell}{2}+1, \ell]} \right\|_{\mathsf{F}}^2 \leq \frac{1}{\ell/2} \|R_\ell - \mathcal{H}_\ell^*\|_{\mathsf{F}}^2.$$

Together with (8) and (7) this gives with probability $\geq 1 - \delta$ that

$$\left\| (\widetilde{F} - F^*) \mathbb{1}_{[\frac{\ell}{2}+1, \ell]} \right\|_{\mathsf{F}} \leq C \left( \sqrt{\frac{\|\Sigma_y\| \, d \, (d_y + d_u + \log \left( \frac{L}{\delta} \right))}{T}} + \sqrt{\frac{\|\Sigma_x\| \, L d_u \log \left( \frac{L d d_u}{\delta} \right)}{T}} \|M_{G^*}\| + \frac{\varepsilon_{\text{trunc}} \sqrt{d}}{\sqrt{\ell}} \right)$$

Replacing $\delta$ by $\frac{\delta}{\log_2 L}$, using a union bound over powers of 2, and summing gives

$$\left\| (\widetilde{F} - F^*) \mathbb{1}_{[1, L]} \right\|_{\mathsf{F}} = O \left( \sqrt{\frac{\|\Sigma_y\| \, d \, (d_y + d_u + \log \left( \frac{L}{\delta} \right)) \log L}{T}} + \sqrt{\frac{\|\Sigma_x\| \, L d d_u \log \left( \frac{L d d_u}{\delta} \right) \log L}{T}} \|M_{G^*}\| + \varepsilon_{\text{trunc}} \sqrt{d} \right).$$

$\square$

*Proof of Theorem 2.2.* We have the bound in Lemma 4.4, and also the same bound for $\left\| (\widetilde{F} - F^*)(0) \right\|_{\mathsf{F}}$ after applying Lemma 4.2 to $(F - F^*)\delta_0$. Finally, note that $\left\| (\widetilde{F} - F^*) \mathbb{1}_{(L, \infty)} \right\|_{\mathsf{F}} = \left\| F^* \mathbb{1}_{(L, \infty)} \right\|_{\mathsf{F}}$ and use the triangle inequality. $\square$

## 5 Experiments

We compared three algorithms for learning the impulse response function: least-squares, and low-rank Hankel SVD with and without the multi-scale repetition. We include details of the experimental setup in Appendix D. Note that to reduce the number of scales, we consider use a slight modification of our Algorithm 1 which triples the size at each iteration instead.

The plots show the error $\left\| F^* \mathbb{1}_{[1, L]} - F \right\|_2$, where $F$ is the estimated impulse response on $[1, L]$, averaged over 10 randomly generated LDS's, as a function of the time $T$ elapsed. We consider systems of order $d = 1, 3, 5, 10$, and memory lengths $L = 27, 81$.

Using SVD significantly reduces the error, supporting our theory which shows that SVD has a "denoising" effect. Additionally, multiscale SVD has better performance than naive SVD when $d$ is moderate, $L$ is large, and data is limited, but the performance is similar in a data-rich setting.

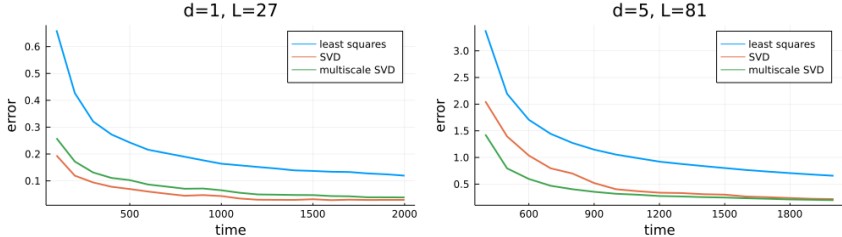

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
