# OpenReview forum: "Improved rates for prediction and identification for partially observed linear dynamical systems"
_NeurIPS.cc/2021/Conference — NeurIPS 2021 Submitted_

### Official Review · Reviewer_ZPKh · 2021-07-11

**Rating:** 5
**Confidence:** 4

**Summary:**

This work studies the estimation of linear dynamical systems with unobserved state. The authors propose a novel algorithm which relies on obtaining low-rank approximations to the Hankel matrix. They provide several results on the performance of their algorithm, including finite time bounds on the error it obtains in estimating the system’s impulse response. As compared to previous work, the main novelty of this paper is eliminating a factor of the horizon in the case where there is no process noise.

**Limitations And Societal Impact:**

The authors do not discuss potential societal impact. However, the work is primarily theoretical so it is difficult to determine what negative societal impacts it may have. Limitations are discussed.

**Main Review:**

Obtaining finite-time bounds for the estimation of linear dynamical systems—in both the setting where the state is observed and the state is unobserved—has been a well-studied problem in the machine learning community over the last several years, and this works fits in that literature. A primary goal of existing works has been to obtain error bounds which do not scale with the “memory” of the system, 1/(1-\rho(A)). While achievable in the case where the state is observed, this has been a more difficult goal to reach in the case when the state is unobserved. Several existing works have addressed this—for example, [1] introduces the notion of "phase rank" to avoid the memory dependence—but it is not clear that the issue has been completely solved (phase rank could be exponential in the dimension in some cases, which is also highly undesirable). This work claims to remove the memory dependence without incurring poor dependence on other quantities, but is only able to do so in the case with 0 process noise.

Pros:
- As noted above, removing the 1/(1-\rho(A)) dependence is an important goal, and as yet has not been completely achieved in the case when the state is unobserved. As such, the problem considered here is relevant and, if solved, would be a contribution.
- The authors do solve the problem in the case when the process noise is 0, eliminating the dependence on the memory in this setting.

Cons:
- The primary issue with this paper is that the memory dependence can only be removed in the case when there is no process noise, i.e. the results improve on the state of the art only when there is no process noise. This is a very restrictive assumption, is likely unrealistic in practice, and it is not clear how interesting this setting is to the learning theory community (existing works have all assumed non-zero process noise).
- Even in the noiseless setting, the memory dependence is not removed for the recovery of the system parameters (i.e. A,B,C,D) but only for the recovery of the impulse response.
- The authors restrict to the case when \rho(A) < 1 while existing works also encompass the case when \rho(A) = 1.
- The algorithm takes as input the memory length of the system, but it is not clear how one should actually set this without knowing the system parameters.
- More care should be taken in comparing to [1]. [1] introduces the idea of phase rank, and provides bounds that scale with the phase rank instead of 1/(1-\rho(A)) in some cases (even with non-zero process noise). No discussion of this is given—it should be clarified what precisely the bounds in [1] work out to be, and what the actual memory dependence suffered is. At minimum the authors should not claim that no existing works obtain bounds that are memory-independent and should show explicit examples where their results improve over the phase rank dependence in [1].

While this work does make a contribution, the restrictive assumptions noted above and the fact that [1] partially solves the problem already are, in my opinion, significant enough that it does not warrant an accept.

[1] Max Simchowitz, Ross Boczar, and Benjamin Recht. Learning linear dynamical systems with semi-parametric least squares. arXiv preprint arXiv:1902.00768, 2019.

**Time Spent Reviewing:**

2

---

> ### Author Response · Authors · 2021-08-10
> **Response**
>
> Thank you for the thorough review and the valuable technical questions you raised. We hope this response addresses your concerns and that you would reconsider your score in light of these explanations. We are happy to respond to follow-up questions.
>
> *"no process noise... is a very restrictive assumption, is likely unrealistic in practice, and it is not clear how interesting this setting is to the learning theory community (existing works have all assumed non-zero process noise)."*
>
> We agree that the result is limited, but we believe the techniques and algorithmic building blocks in this work to be useful (or even pre-requisite) in designing and analyzing algorithms in the general case of non-zero process noise. In particular, our result hinges on the "de-noising" effect of SVD, and has the potential to be extended to more complex subspace identification algorithms that address process noise, such as N4SID. The special case of zero process noise already includes nontrivial technical challenges. We emphasize that our technical contributions are of independent interest: Lemma A.1/Theorem A.2 give a concentration bound for the sample covariance matrix for correlated inputs and requires careful use of concentration bounds in the Fourier domain; we use this to obtain high-probability error bounds for estimation of a transfer function in $\mathcal H_\infty$ norm from a single rollout. This improves on or generalizes related bounds in the literature [OO19, DMR19].
>
> *Even in the noiseless setting, the memory dependence is not removed for the recovery of the system parameters (i.e. A,B,C,D) but only for the recovery of the impulse response.*
>
> This is a good point - we leave it to future work to improve the bound for recovery of the system parameters or to prove a lower bound.
>
> *The authors restrict to the case when \rho(A) < 1 while existing works also encompass the case when \rho(A) = 1.*
>
> Strictly speaking, with our problem formulation, an offline result for $\rho(A)=1$ is not possible, in the following sense: a perturbation may affect finite-time horizon predictions by an arbitrarily small amount while affecting unbounded change in $\mathcal H_\infty$ norm. For $\rho(A)=1$, there are two possible ways to obtain a result:
>
> 1. Only require that we learn $[D, CB, CAB,\ldots, CA^{L-1}B]$ up to a given $L$. If we are given multiple trajectories of length $O(L)$, this is achievable with our techniques using a similar analysis, with the finite rollout length taking the place of the memory length in the analysis. This is the problem considered in Simchowitz et al. 2018, though they are able to accomplish this with a single trajectory.
>
> 2. Achieve an online result. This is a more challenging problem, and is a good direction for future work.
>
> *The algorithm takes as input the memory length of the system, but it is not clear how one should actually set this without knowing the system parameters.*
>
> We wrote the result with known memory length (and $\epsilon_{\text{trunc}}$) for simplicity. A standard technique to convert the algorithm to one that achieves the correct rates as $T\to \infty$ without knowing the memory length is to use the "doubling trick": increase the memory length by a constant every time the number of timesteps doubles, so that the memory length scales as $\log_2(T)$. This works because the impulse response decays exponentially. See Algorithm 1 in [TP20] for an example where this is carried out.
>
> [TP20] Tsiamis and Pappas, Online Learning of the Kalman Filter with Logarithmic Regret
>
> *"More care should be taken in comparing to [1]. [1] introduces the idea of phase rank, and provides bounds that scale with the phase rank..."*
>
> [1] https://arxiv.org/pdf/1902.00768.pdf
>
> It is true that [1] works under more general conditions of noise and marginal stability - and is clearly a landmark paper in this regard. However, its focus is somewhat different, so it does not recover the same guarantees in our problem setting - specifically our rates for recovery of Markov parameters in $\mathcal H_2$ norm.
>
> First, we note that the main bound in [1] is for the *truncated* Markov parameters $G=[D, CB,\ldots, CA^{T-2}B]$ (Proposition 2.1/Theorem 3.3), and they in fact incur dependence on T in their rates (not to be confused with T in our notation). If we wanted to capture all but $\epsilon$ of the impulse response, then we would have to take T to be the memory length (L in our notation), and this would in fact incur the undesirable memory dependence. The advantage is that they are able to achieve consistent recovery of system parameters, for which it is not necessary to take T to be as large as the memory length.
>
> Secondly, we note that low phase rank is a restrictive assumption which we do not expect to hold generically for "random" linear dynamical systems with eigenvalues close to 1 - rather, it requires special structure on the eigenvalues. To construct a class of random dynamical systems with large phase rank, consider $A=rO$ where $1-1/L^2 \le r\le 1$ and $O$ is an orthogonal matrix with eigenvalues $e^{i\theta_k}$ randomly chosen on the unit circle (in complex conjugate pairs). Letting $d$ be the dimension of the latent state $x_t$, for $L\ge d$, the $(\alpha, L)$-phase rank (for constant $\alpha$) can be lower bounded in terms of the number of distinct buckets occupied by $\lbrace L\theta/(2\pi)\rbrace$, where $\lbrace\cdot\rbrace$ denotes fractional part, and the buckets are $[0,1/L), [1/L,2/L),\ldots [(L-1)/L,1)$.
> (Recall that phase rank essentially groups eigenvalues that differ by close to a factor of a $L$th root of unity.)
> With high probability, $\Omega(d)$ buckets are occupied. As the result of Simchowitz et al. is exponential in the phase rank, their paper only gives an $\exp(cd)$ upper bound in this case. Thus, under our problem setting, our algorithm performs better for this class of LTI systems.
>
> We will add details of this comparison to the appendix.

---

> ### Comment · Reviewer_ZPKh · 2021-08-23
> **Updated Review**
>
> After reading the reviewers’ comments, I am willing to raise my score to a 5, as I believe the comparison with [1] is satisfactory. That the main novelty of this work (removing memory dependence) only holds when there is no process noise is still a major shortcoming to me and prevents me from raising my score any more. For future revisions of this work, I would suggest making the following changes, which could greatly strengthen the paper:
>
> - Add a lower bound showing that the memory dependence is in some cases necessary when there is process noise. Intuitively, it seems like this could be the case, as the memory is roughly the amount of time a single noise sample will affect the state. If one could show this, it would greatly strengthen the results here by showing they are effectively unimprovable, and really present a full story on memory dependence in system identification rates.
> - Along the same lines, it would be helpful to show through an example that existing algorithms suffer this memory dependence even when there is no process noise. I believe this is less necessary than my first suggestion, but would help justify the new algorithm proposed here.

---

> > ### Author Response · Authors · 2021-09-02
> > **Response to updated review**
> >
> > We are glad that the reviewer finds the comparison satisfactory. We will certainly take into account the suggested improvements in the next revision.

---

### Official Review · Reviewer_iHne · 2021-07-16

**Rating:** 7
**Confidence:** 3

**Summary:**

This work considers the problem of learning linear dynamical systems with partial observability both in terms of learning the impulse response and the system parameters upto similarity transformation (using Ho-Kalman algorithm). The main technical contribution is Algorithm 1 and Theorem 2.2 which presents the error bounds for Algorithm 1. This shows that the impulse response can be learned upto frobenius error $\sqrt{d/T}$, without any dependence on the mixing time/memory length of the dynamics. The main technical tool used in Algorithm 1 is the least squares estimation of the Hankel matrix at various scales to cover dependences from inputs at various time instants and denoising it with SVD since the true Hankel matrix is constrained to have rank d.

The experiments illustrate the advantage of the algorithm compared to least squares regression.

**Limitations And Societal Impact:**

Yes.

**Main Review:**

I think the algorithm and its analysis is novel on both technical and conceptual level. It solves an important problem with near optimal bounds which does not depend on the memory length by using the elegant idea of SVD to denoise the Hankel matrix estimate obtained through least squares regression. Rest of the algorithm seems to be fairly standard.

I think this deserves to be published in NeurIPS 2021.

Some comments:

1. The improved rates only hold when there is no additional noise in the hidden dynamics of the system. (This is mentioned in the paper)
2. The parameter recovery in Theorem 2.3 has an additional factor L, which is not clearly explained.
3. It would be great if the magnitude of $\epsilon_{\mathsf{trunc}}$ is mentioned.
4. It would be useful to mention the computational complexity of the algorithm.

**Time Spent Reviewing:**

4 hours

---

> ### Author Response · Authors · 2021-08-10
> **Response**
>
> Thank you for the encouraging review. Space permitting, we will incorporate your suggestions in the revision.
>
> *The parameter recovery in Theorem 2.3 has an additional factor L, which is not clearly explained.*
>
> We speculate that the factor of L comes from recovering system parameters from the SVD at a single scale. The challenge with using a multiscale approach here is that estimates of system parameters at different scales must be collated. We leave the possibility of improving this bound to future work.
>
> *"It would be great if the magnitude of $\epsilon_{\text{trunc}}$ is mentioned."*
>
> Commonly, one assumes that $||CA^tB||\le M\rho^{t}$ for some $M$ and $\rho$ (greater than or equal to the spectral radius of $A$). Then in the noiseless case, $\epsilon_{\text{trunc}}\le M\frac{\rho^{2L-1}}{1-\rho}\sqrt{d_u}$. See the response to reviewer ZPKh for a way to modify the algorithm when these parameters are not known.
>
> *It would be useful to mention the computational complexity of the algorithm.*
>
> The computational complexity of the algorithm (for $T\gg d$) is determined by the complexity for the limiting step - solving the least squares problem for F - which takes time $O((Ld_u)^3 + LT^2d_ud_y) = O(LT^2d_ud_y)$ (with improvements if using fast matrix multiplication).

---

> > ### Comment · Reviewer_iHne · 2021-09-10
> > **Thanks for the Response**
> >
> > I thank the authors for the response. After reading the reviews and responses, I have decided to keep my score the same.

---

### Official Review · Reviewer_BcJQ · 2021-07-17

**Rating:** 6
**Confidence:** 4

**Summary:**

The paper proposes an algorithm for learning LTI systems with zero process noise. Given an input trajectory generated from independent Gaussian input, the algorithm estimates the impulse response function, which in turn can be used to estimate system parameters up-to similarity transformation.

The key idea is the multi-scale smoothing of the impulse response: A first estimate is obtained using linear regression. Denoising of the response function estimate is then performed by low-rank approximation of Hankel matrices of different sizes, with the insight being that the size of the matrix determines which parts of the response functions are preserved under the low-rank approximation.

The proposed algorithm achieves an error rate that scales only logarithmically with the system memory. The efficacy of the algorithm is demonstrated on a synthetic problem.

**Limitations And Societal Impact:**

I do not see a potential negative societal impact in the paper.

**Main Review:**

Overall, this is a well-written paper. The contribution is clear and the flow of ideas is logical.

My main concern is that the experiments were conducted exclusively on synthetic data. While It is beneficial to have control over the parameters of the problem. It is equally important to demonstrate the efficacy of the algorithm in real-world data.

Other comments:
* L96-98 gives the impression that Frobenius norm lower bounds the operator norm whereas the opposite should be true.
* L130: It is not clear what "diagonals that are $\Theta(l)$" means. Also, do you mean skew-diagonals?
* L137: What is G? Also, it would be good to define the indicator function.
* Algorithm 1 - Line 4: $(F * u)(t)$  (use parentheses)
* L146: $\Sigma_x$ instead of $\Sigma_y$ for process noise.

**Time Spent Reviewing:**

3

---

> ### Author Response · Authors · 2021-08-10
> **Response**
>
> Thank you for supporting the paper, catching the typos, and suggesting the clarifications. We will fix the issues in the revision.
>
> *"experiments were conducted exclusively on synthetic data."*
>
> We agree that this would be good to do in future work. One barrier to utilizing the current algorithm in practice is that its improved guarantees hold only in the case of zero or small process noise. Nevertheless, we believe our work to be a valuable mathematical contribution, and a potential building block for algorithms in a more general setting.
>
> We believe our technical contributions are of independent interest: Lemma A.1/Theorem A.2 give a concentration bound for the sample covariance matrix for correlated inputs and requires careful use of concentration bounds in the Fourier domain; we use this to obtain high-probability error bounds for estimation of a transfer function in $\mathcal H_\infty$ norm from a single rollout. This improves on or generalizes related bounds in the literature [OO19, DMR19].
>
> *L96-98 gives the impression that Frobenius norm lower bounds the operator norm whereas the opposite should be true.*
>
> We will clarify that there is an implicit factor of $d$ scaling between $\mathcal H_2$ and $\mathcal H_\infty$ here.
>
> *It is not clear what "diagonals that are $\Theta(l)" means. Also, do you mean skew-diagonals?*
>
> You are right, it should be "skew-diagonals". For an $\ell\times \ell$ matrix $H$, we mean the skew-diagonals consisting of entries $H_{ij}$ with $c_1\ell \le i+j \le c_2\ell$.
>
> *What is G?*
>
> $G^*$ is defined by $G^*(t) = CA^{t}$ (the impulse response for the process noise). We will add this.

---

### Author Response · Authors · 2021-08-10
**General response**

We thank the reviewers for their detailed reading of the paper and their insightful comments. We address reviewer suggestions, concerns, and questions in individual comments below.

---

### Decision · Program_Chairs · 2021-09-27

**Decision:**

Reject

**Comment:**

The paper studies the problem of learning a partially observed linear dynamic system and provides non-asymptotic system identification guarantees that scale with the dimensionality of the system, rather than the memory length. This is definitely an interesting direction, but the reviewers identified several limitations with the results in the current paper. Most notably is the fact that the paper only concerns systems with no process noise.

I am sympathetic to the fact that making progress on a hard problem can take some time and may require making assumptions to obtain preliminary results and develop new ideas. However, it would be worthwhile to explore the consequences of the assumptions more to provide a more complete story. In the context of the present paper, as suggested by the reviewer, it would help to understand if other algorithms have a memory dependence in the no-process-noise regime and also if there are lower bounds in the more general setting. Such auxiliary results would complete the picture and make the paper much stronger.

Thus, we recommend rejecting the paper at this time.